



# Teleconnection of Sahara to Mediterranean Basin evidenced by consecutive dust storms

Vincenzo Carbone[1,3,✠], Vincenzo Capparelli[1], Jessica Castagna[2], Fabio Lepreti[1,3], Alfonso Senatore[2], and Giuseppe Mendicino[2]

[1]Department of Physics, University of Calabria, Ponte P. Bucci 31C, 87036 Rende, CS, Italy
[2]Department of Environmental Engineering, University of Calabria, Via Pietro Bucci, Rende, 87036, Italy
[3]National Institute for Astrophysics, Scientific Directorate, Viale del Parco Mellini 84, 00136 Roma, RM, Italy
[✠]Deceased 21 January 2025

**Correspondence:** Fabio Lepreti (fabio.lepreti@unical.it)

**Abstract.** Saharan Dust Outbreaks frequently hit the Mediterranean Basin, lasting for a few days. These phenomena have various implications for the ecosystem of the entire basin, affecting the atmosphere, lithosphere, biosphere, hydrosphere, and cryosphere. Moreover, they cause numerous hazards to human society, especially concerning the environment and health, and are particularly significant to people living in a "Dust Belt" around Sahara, including nearby areas such as the Mediterranean Basin. This study demonstrates that continuous dust intrusions from the Sahara, transported across distant geographic regions, cannot be considered random events; rather, they show long-range correlations for timescales shorter than 80 days. This behaviour generates a persistent and recurrent atmospheric pattern at inter-annual time scales and synoptic spatial scales, thus opening a new perspective for climate studies and evidencing a new kind of teleconnection between North Africa and the Mediterranean Basin.

## 1 Introduction

The desert dust, caused by strong wind erosion over dry land, is the most abundant source of particulate matter in the atmosphere and is estimated by models to range between 6 to 30 Tg/year (Kok et al., 2017). The study of dust storms from deserts, their origin, transport, and deposition is a subject of growing importance and interest in the study of global environmental changes, because dust storms have great significance for both the physical environment, such as the climate effects and Earth's global energy balance, and human society, producing health effects (i.e., mortality, morbidity, and other effects as respiratory and cardiovascular disorders), transport accidents due to poor visibility during desert storms, and issues on electricity generation and distribution (Goudie and Middleton, 2006; Middleton, 2017; Kok et al., 2021).

Dust alters climate by direct and indirect interaction with incoming solar and outgoing long-wave radiation (Rosenfeld et al., 2008; Adebiyi et al., 2023). Regarding the direct effect, dust in the atmosphere can absorb and scatter solar radiation altering the Earth's global energy balance. The dust direct radiative effect ranges between -0.48 and +0.20 W m⁻² depending on the size distribution, because fine dust particles mainly produce scattering of radiation, cooling the Earth, meanwhile, coarse dust absorbs radiation causing a warming (Kok et al., 2021). As an example, a very intense desert dust event, i.e. the June 2020





historical Saharan Dust storm, that blew approximately 8 Tg over the eastern tropical Atlantic, produced a +14 W m$^{-2}$ increase in the surface net radiation flux at night, an increase of 1.1 K of the ocean surface temperature and a 1.8 K warming of the air temperature (Francis et al., 2022c). Moreover, as an indirect effect, aerosol particles alter the cloud condensation nuclei (CCN), changing cloud droplets' size distribution and altering the rainfall processes (Bryson and Baerreis, 1967; Maley, 1982; Rosenfeld et al., 2008). Furthermore, dust increases mineral nutrient supply to the oceans (Gherboudj et al., 2017).

Dust event formation is influenced by the major climate indices, such as the Atlantic Multi-decadal Oscillation (AMO) which influences the global dust load (Shao et al., 2013), the North Atlantic Oscillation (NAO), a climate forcing for dust storm spreading from the Mediterranean region until China (Li et al., 2023; Sabatier et al., 2020), the Pacific Decadal Oscillation influencing desert dust in Asia (Huo et al., 2023), the El-Niño Southern Oscillation (ENSO) that is the driver for Australian dust events (Shao et al., 2013), whereas La-Niña affects the South American events (Shao et al., 2013). Moreover, the dust storms are influenced also by drought phases, such as during the Australian "Millennium Drought", (McTainsh et al., 2005; O'Loingsigh et al., 2017), and land-cover alterations (Kandakji et al., 2021; Ebrahimi-Khusfi and Soleimani Sardoo, 2021). Therefore, monitoring dust storms can be indicative of environmental change.

Among the deserts worldwide (i.e., in Asia, Northern America, and the Southern Hemisphere), the Sahara desert, placed in the "Dust Belt" that expands through North Africa, the Middle East, and China (Dominguez-Rodriguez et al., 2020), provides the largest amount of desert dust in the atmosphere, contributing to approximately 50% of the total dust (Kok et al., 2021). The Saharan Dust storms are a common impulsive phenomenon in the Mediterranean area (Bencardino et al., 2019; Bibi et al., 2020; Salvador et al., 2022), until northern Europe (Goudie and Middleton, 2006), reaching even the American continent (Francis et al., 2022b). The drivers of dust storms are atmospheric conditions, which uplift the dust and can transport it for a long-range (Francis et al., 2020, 2022a; Littmann, 1989). This is a phenomenon that is observable not only in the industrial epoch but it is evidenced also in paleoclimatic proxies (Clifford et al., 2019) in the last 2000 years.

Due to the growing interest in the environmental and climatic consequences of phenomena of this type, relatively long time series of relevant measurements are now routinely acquired. Therefore, it has become of particular interest to study in detail the statistical properties of these time series and to interpret them in connection with the atmospheric circulation patterns occurring in the areas of interest. In this paper, using PM10 measurements that are acquired throughout Europe, we look at the phenomenon as a point process obtained by investigating the time evolution of the PM10 content at several locations in central and southern Europe, where the phenomenon occurs more frequently. The basic statistical behaviour of the point process provides information about some fundamental properties of the dust storms' occurrence, especially about the long-range time correlations produced by the atmospheric circulation dynamics.

## 2   Data

We performed the statistical analysis on daily PM10 data provided by the European Environment Agency (EEA), which collects the environment and climate data of 32 member countries and 6 cooperating countries (URL:https://discomap.eea.europa.eu/map/fme/AirQualityExport.htm). The main methods accepted by the EEA for the PM10 collection are gravimetric



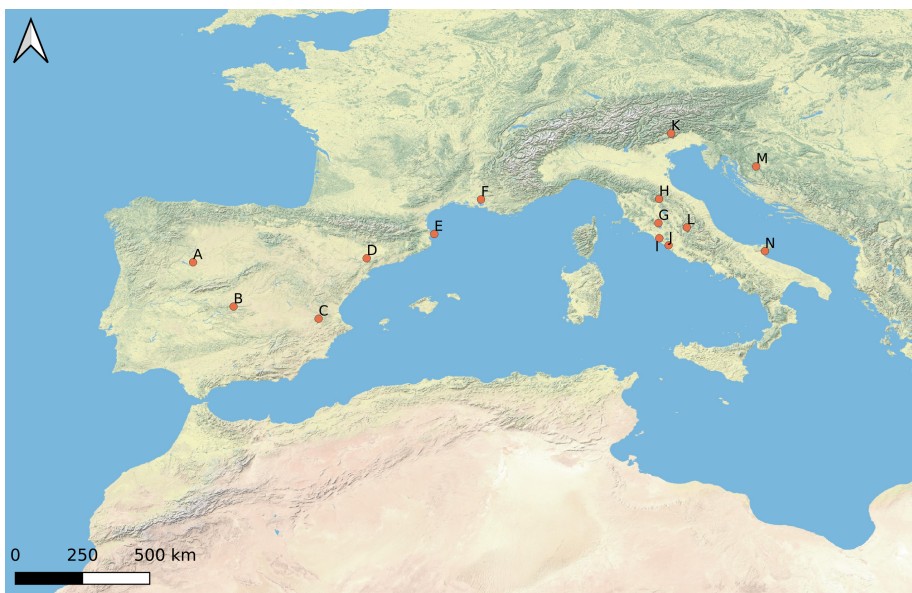

**Figure 1.** The locations of the stations used in the analysis are indicated by the red dots. Labels refer to the stations IDs in Table 2. Credits: ESRI, U.S. National Park Service.

(daily sampling), TEOM, and Beta ray attenuation (both hourly sampling), while the required time resolution for the dataset is daily (Wim Mol, 2013).

To focus on the natural source of extreme dust events, limiting the anthropic influence on the PM10 monitoring, we selected only "background" stations placed in rural, rural regional, or at least urban areas (EU, 2005, 2008, 2011), where typically the natural events as the SDO are easily identifiable because the other anthropic sources can be considered negligible (Bencardino et al., 2019; Castagna et al., 2021b; Petroselli et al., 2024). Several stations show numerous data gaps in the PM10 time series, hence to prevent the results distortion of our analysis, a strict selection of stations was made. The selection criteria were: (i) a signal with at least 10 years of data (even if not consecutive); (ii) a percentage of gaps less than 10% of the total data. Following this selection, we analysed 14 stations, with a uniform distribution over the western Mediterranean basin, as shown in Figure 1. Further information on stations, such as the site's coordinates, elevation, country, and data period can be found in Table 1. Unfortunately, there were no stations satisfying the aforementioned criteria that were located in the eastern Mediterranean region, limiting this study to the western part.

We downloaded PM10 time series of the selected stations using the library "saqgetr" in R software. Overall the period starts from the 1st of January 2002 until to 31st of December of 2023.





**Table 1.** Description of the EEA PM10 stations: ID as reported in Fig. 1, EEA Site Id, geographic coordinates (Latitude and Longitude), Elevation, Country, Site Type and Area, Monitoring Date Start and Date End.

| ID | Site Id | Lat (°N) | Lon (°E) | Elev. (m a.s.l.) | Country | Site Type | Site Area | Date Start | Date End |
|----|---------|----------|----------|------------------|---------|-----------|-----------|------------|----------|
| A | es0013r | 41.23889 | -5.8975 | 985 | Spain | background | rural | 2002-01-01 | 2021-12-31 |
| B | es0001r | 39.54694 | -4.35056 | 917 | Spain | background | rural | 2008-01-01 | 2021-12-31 |
| C | es0012r | 39.08278 | -1.10111 | 885 | Spain | background | rural | 2002-01-01 | 2021-12-31 |
| D | es0014r | 41.39389 | 0.73472 | 470 | Spain | background | rural | 2002-01-01 | 2021-12-31 |
| E | es0010r | 42.31917 | 3.31583 | 76 | Spain | background | rural | 2002-01-01 | 2021-12-31 |
| F | fr0041 | 43.639 | 5.101097 | 82 | France | background | urban | 2013-01-01 | 2023-12-31 |
| G | it2023a | 42.73657 | 11.87643 | 436 | Italy | background | rural | 2011-01-01 | 2023-12-29 |
| H | it1681a | 43.66028 | 11.90167 | 650 | Italy | background | rural | 2008-12-01 | 2023-12-29 |
| I | it0884a | 42.15786 | 11.908954 | 532 | Italy | background | rural | 2008-11-02 | 2023-12-29 |
| J | it0952a | 41.88945 | 12.266327 | 66 | Italy | background | rural | 2010-07-27 | 2023-12-29 |
| K | it1790a | 46.1625 | 12.360833 | 615 | Italy | background | rural | 2008-01-01 | 2023-12-28 |
| L | it0989a | 42.57259 | 12.962063 | 948 | Italy | background | rural | 2008-07-17 | 2023-12-29 |
| M | hr0013a | 44.89933 | 15.60978 | 0 | Croatia | background | rural regional | 2013-12-31 | 2023-12-31 |
| N | it1601a | 41.66528 | 15.945 | 30 | Italy | background | rural | 2009-06-01 | 2023-12-29 |

## 3 Methods

### 3.1 Identification of Saharan Dust events

From the measurements of PM10 at a given $j$-th location, we built up a time series $PM10_j(t)$, and we selected (Johnston et al., 2011; Calidonna et al., 2020) the most severe PM events when $S_j(t) \geq \langle PM10_j(t)\rangle + 2\sigma_j$, where $\sigma_j$ was the standard deviation of the signal $PM10_j(t)$. For each day with PM10 exceeding the threshold value, the Saharan Dust intrusion was confirmed considering the dust influence over the station studying the back-trajectories of the monitored air masses with the numerical models and satellite images, which is a method largely applied in literature (Escudero et al., 2007; Bencardino et al., 2019; Russo et al., 2020; Castagna et al., 2021a; Barnaba et al., 2022). In particular, the dust presence in the troposphere was confirmed by the NAAPS (Navy Aerosol Analysis and Prediction System) Global Aerosol Model maps (http://www.nrlmry.navy.mil/aerosol) integrated with the 96 h back-trajectories calculated at three arrival heights (500, 1000, 1500 m a.s.l) by HYSPLIT (Hybrid Single-Particle Lagrangian Integrated Trajectory) model driven by Reanalysis set as meteorological conditions (https://www.ready.noaa.gov/hypub-bin/trajasrc.pl). In particular, for the PM10 peaks, if the 96 h backtrajectories highlighted that the airmasses encountered along the way the Saharan dust, we categorized that event as Saharan. In this way, we ensured that further statistics analyses were performed only on 1652 days, classified as the Saharan Dust days.





## 3.2 Synoptic meteorological patterns

The synoptic meteorological conditions were evaluated to establish the atmospheric circulation that favours the SDO. We downloaded the daily Geopotential height fields at the 850 hPa level at 12 UTC from the NCEP/NCAR Reanalysis dataset and performed a non-hierarchical k-means cluster analysis. The optimal number of clusters was established by applying a heuristic technique, the Elbow Method, based on the graphic observation of the behaviour of the Within-Cluster Sum of Squares (WCSS) values in function of the number of clusters. The optimal number of clusters for the cluster analysis corresponds to an "elbow"

point when although the possible number of clusters is increased the WCSS doesn't decrease significantly suggesting an over-fitting. To assess the seasonal atmospheric circulations and their relationship with the dust intrusion, we defined:

- the Frequency of Dust in Cluster ($f_{DC}$): given by the ratio between the yearly SDO events with a cluster and the total number of occurrences of the cluster.

- the Seasonal Cluster Frequency ($f_{SC}$): given by the ratio between the seasonal days with a cluster and the total number

of days in that season;

- the Seasonal Frequency of Dust in Cluster ($f_{SDC}$): given by the ratio between the seasonal SDO events with a cluster and the seasonal total number of occurrences of the cluster.

The seasons were considered in the following way: Winter (Jan-Mar), Spring (Apr-Jun), Summer (Jul-Sep), and Autumn (Oct-Dec).

## 4 Results

### 4.1 Atmospheric circulation patterns during Saharan Dust events

At first, to better interpret the memory times of Saharan dust days resulting from statistical analysis, we studied the physical mechanisms that favour Saharan dust transport towards the Mediterranean area from the perspective of seasonal atmospheric circulations. Indeed, the Saharan Dust Outbreaks (SDO) are favoured by climate drivers triggered by seasonal atmospheric

circulations. Considering the investigated period 2002-2023 and the study area, i.e., the western Mediterranean Basin, the SDO are more frequent during the summer (29%), followed by winter (28%) and spring (26%). The atmospheric circulations blowing the Saharan Dust are identified using the k-means cluster analysis for the 850 hPa geopotential height. The non-hierarchical k-means analysis for the 850 hPa geopotential height at 12:00 UTC highlights 4 circulation scenarios (Fig. 2). In Cluster 1 (Fig. 2a), an anticyclone is over the western Mediterranean Basin, including Spain, France, Italy, and the Balkans,

with a high-pressure center in the Atlantic Ocean. The synoptic conditions in Cluster 2 (Fig. 2b) show the Azores current with high pressure in the Atlantic Ocean and lower pressure over the western Mediterranean area. Cluster 3 (Fig. 2c) describes an Arctic cyclone moving toward Europe, while Cluster 4 (Fig. 2d) shows that the anticyclonic circulation is moving from Northern Africa toward the Mediterranean Basin.





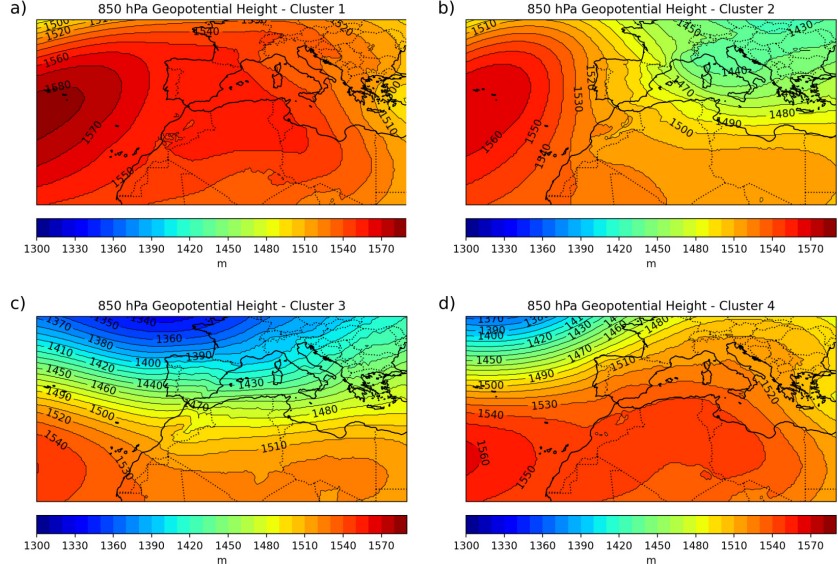

**Figure 2.** Composite synoptic maps of 850 hPa geopotential height at 12 UTC obtained by the K-means clustering.

Generally, the atmospheric conditions favouring the uplift and the transport of the Saharan Dust are represented by Cluster
2, where the high pressure starting from Africa reaches Spain overflying the Atlantic Ocean, followed by Cluster 4 when winds
enriched by the uplifted Saharan Dust blow from Northern Africa. Indeed, the Frequency of Dust in Cluster (see Methods
paragraph) $f_{DC}$ is equal to 17% and 14%, respectively for Cluster 2 and Cluster 4.

Moreover, we analysed the SDO seasonality by studying the prevalent atmospheric circulations during the Saharan Dust
events. During the winter, Cluster 1 represents the prevalent atmospheric conditions with a Seasonal Cluster Frequency $f_{SC}$
(see Methods paragraph) equal to 34%, but this atmospheric pattern does not favour particularly the SDO. On the other hand,
the atmospheric conditions favouring the transport of the Saharan Dust in winter are described by Cluster 2 and Cluster 4,
which in winter are less frequent than Cluster 1 (with a $f_{SC}$ of 23% and 22%, respectively). Indeed, in winter, we found a
Seasonal Frequency of Dust in Cluster $f_{SDC}$ (see Methods paragraph) equal to 20% of events for both Cluster 2 and 4, which
is the largest number of events in frequency terms. In spring the most frequent atmospheric pattern is Cluster 2 ($f_{SC}$ equal to
32%), which induces a large number of SDO events ($f_{SDC}$ equal to 20%, as for Cluster 2 and Cluster 4 in winter) followed by
Cluster 4 ($f_{SC}$ equal to 29%) favouring a high number of SDO occurrences ($f_{SDC}$ of 13%). The summer season represents a
breaking point with the temporal distribution of atmospheric patterns because a net dominance of Cluster 2 with a $f_{SC}$ of 72%
occurs. Moreover, the SDO occurrences with Cluster 2 conditions are confirmed also for summer ($f_{SDC}$ of 17%). Finally, in
autumn the main circulation pattern is represented by Cluster 4 ($f_{SC}$ of 33%), which is the cluster that most favours SDO in
this season ($f_{SDC}$ of 13%).





## 4.2 Results on the statistics of persistence times between dust storms

As a statistical proxy, we investigate the basic statistics of persistence times between two consecutive events, defined as $\tau = t_{i+1} - t_i$, being $t_i$ the starting time of an event, that is the first value when $S_j(t_i)$ overcome the threshold. The statistics of persistence times between two consecutive events, also called waiting times, is a point process that provides information on the

possible modelling of the phenomenon underlining a sequence of impulsive events. In particular, it can be easily shown that, if $\lambda$ is the rate of occurrence of the impulsive events, assumed to be constant, then the distribution function of persistence times of completely uncorrelated events is an exponential (Gardiner et al., 1985) $f(\tau \geq \Delta t) = \lambda^{-1} \exp(-\lambda \Delta t)$. This represents a classical memoryless Poisson process. Any distribution that significantly deviates from the exponential function describes a process which is characterized, to some extent, by the presence of correlations, that is, a memory process.

In Figure 3 we report some examples of cumulative distribution functions (CDFs) of persistence times $f(\tau \geq \Delta t)$ as a function of $\Delta t$. It is evident that for low values of $\Delta t$ the distribution is strongly different from a memoryless exponential distribution, rather it is described by a well-defined power law. The power law is robust and describes all the data sets we examined. A simple fit on the data, in the range $5 \leq \Delta t \leq 50$ days, gives the resulting CDF

$$f(\tau \geq \Delta t) = \left( \frac{\Delta t}{\Delta t_0} \right)^{-\mu} \tag{1}$$

with different values of $1.9 \leq \Delta t_0 \leq 4.1$ days, and of the scaling exponent $0.8 \leq \mu \leq 1.1$, very close to $\mu \simeq 1$. This is a rather interesting and completely novel result on Saharan Dust storms, showing that consecutive Saharan Dust events are correlated. In other words, the phenomenon which drives dust clouds on short inter-seasonal times cannot be considered as a Poisson memoryless phenomenon, rather stochastic consecutive dust cloud events are strictly correlated in time.

For persistence times greater than $50 \div 80$ days, depending on the sample, we observe an abrupt cut-off of the CDF, showing

that on longer times the correlation between dust storms is completely lost. To describe the whole CDF we can use the same power law distribution (1) with an exponential cut-off at large persistence times, namely

$$f(\tau \geq \Delta t) = \left( \frac{\Delta t}{\Delta t_0} \right)^{-\mu} e^{-\gamma \Delta t} \tag{2}$$

Since the characteristic parameters of the distribution vary weakly from one location to another, in order to have better statistics, we collect all the persistence times $\Delta t$ regardless of the location, thus generating a unique catalogue. In Figure 4 we report

the CDF of the whole catalogue, which represents, therefore, the statistical behaviour of the dust storm phenomena. The data for $\Delta t \leq 50$ days can be fitted with the power law 1 (red line on Figure 4) with characteristic parameters $\Delta t_0 = (2.932 \pm 0.082)$ days and $\mu = 0.942 \pm 0.011$. The whole CDF can be fitted using equation 2 (blue line on Figure 4) with characteristic parameters: $\mu = 0.757 \pm 0.003$, $\Delta t_0 = (2.378 \pm 0.014)$ days and $\gamma = (84.1 \pm 0.6) \times 10^{-4}$ days$^{-1}$, corresponding to a decay time of about $\gamma^{-1} \simeq 118$ days.

In order to find a physical interpretation to the results just shown, the same analysis is reproduced for the entire dataset but subdividing the waiting times into different seasons. The results are shown in Figure 5, where it can be seen that the only



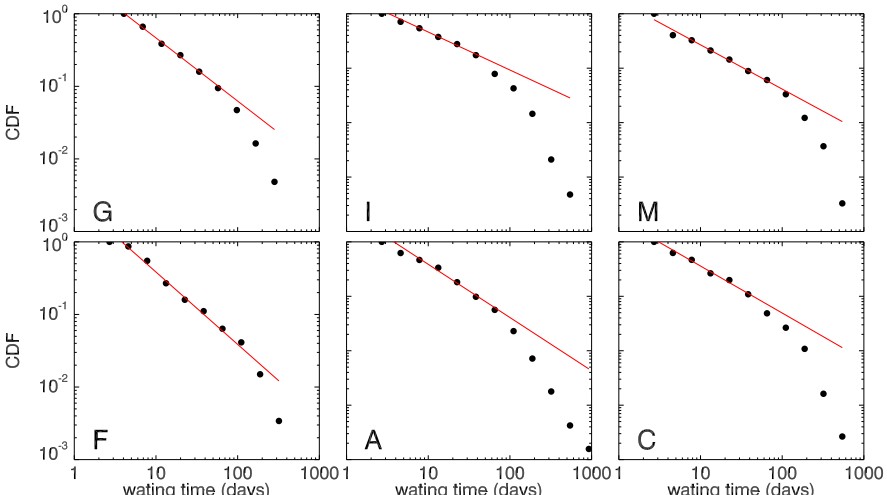

**Figure 3.** Cumulative Distribution Function (CDF) for a few stations highlighted in fig. 1. The red lines show the power law fits with the function (1) over the range $5 \leq \Delta t \leq 50$ days. The letters indicate the corresponding ID station (see Figure 1 and Table 1).

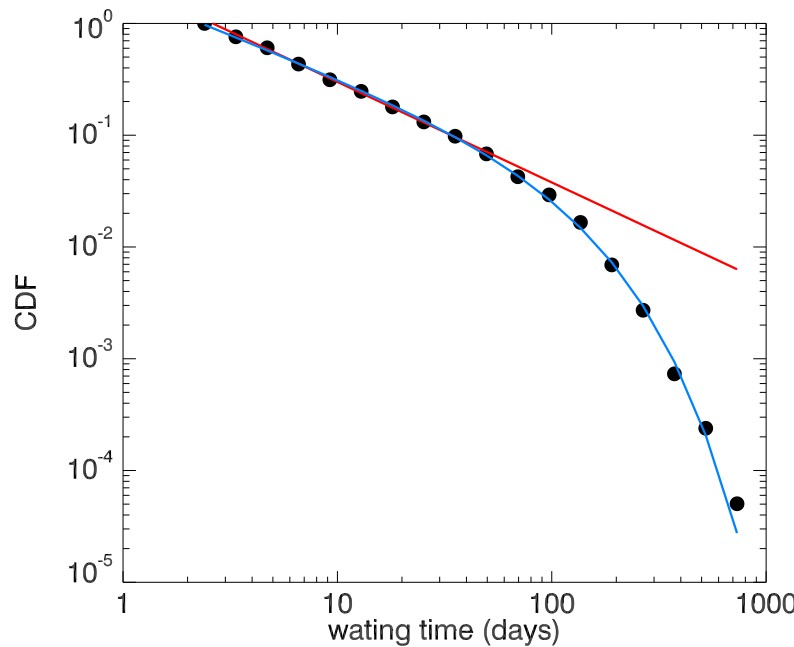

**Figure 4.** Cumulative Distribution Function (CDF) for the whole dataset of persistence times regardless the geographic location of the stations. The red lines show the power law fits with the function (1) over the range $5 \leq \Delta t \leq 50$ days. The blue lines represent the fit with the function (2).




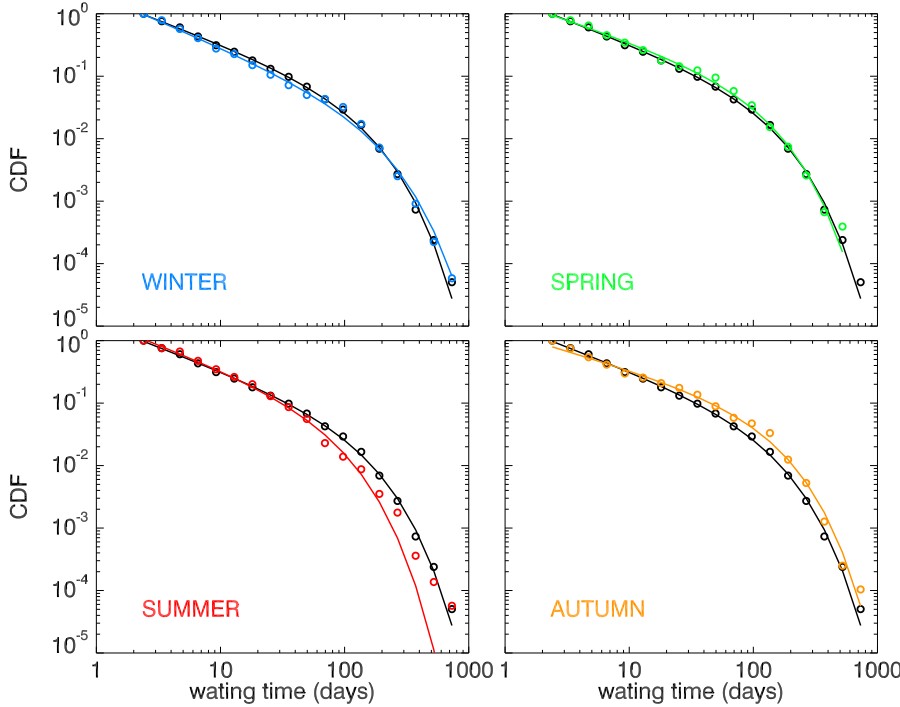

**Figure 5.** Cumulative distribution function (CDF) for the entire persistence time data set, irrespective of the geographical location of the stations (black line). The different colours show the Cumulative Distribution Function (CDF) divided into seasons: Winter (blue), Spring (green), Summer (red), Autumn (orange).

season that differs slightly from the annual trend is summer (red line), showing a different trend for waiting times greater than 90 days with a marked discrepancy between the CDF and function 2 resulting in $\gamma$ parameter from fit and related error (see Table 2).

| Time interval | $\Delta t_0$ (days) | $\mu$ | $\gamma$ (days$^{-1}$) |
|---|---|---|---|
| Annual | $2.378 \pm 0.014$ | $0.757 \pm 0.003$ | $(84.2 \pm 0.6) \times 10^{-4}$ |
| Winter | $2.374 \pm 0.026$ | $0.850 \pm 0.005$ | $(65.6 \pm 1.1) \times 10^{-4}$ |
| Spring | $2.296 \pm 0.028$ | $0.675 \pm 0.005$ | $(97.9 \pm 1.1) \times 10^{-4}$ |
| Summer | $2.689 \pm 0.027$ | $0.769 \pm 0.007$ | $(140.4 \pm 2.7) \times 10^{-4}$ |
| Autumn | $2.374 \pm 0.026$ | $0.850 \pm 0.005$ | $(85.3 \pm 1.1) \times 10^{-4}$ |

**Table 2.** The characteristic parameters obtained from the fit of the function 2 on the cumulative distribution functions (CDFs) at different time intervals, as illustrated in Figure 5.





### 4.3 A local Poisson hypothesis

As written above, the seasonal dynamics of dust storms suggest a link with the atmospheric circulation patterns. This means that the rate of cloud transport, on times longer than one year, varies locally and cannot be considered as constant. In this case, a local Poisson hypothesis changes the waiting time probability of occurrence. In fact, by conjecturing that the phenomenon is Poissonian, and the rate of occurrence of dust clouds changes locally according to a given probability of occurrence, it can be shown (Wheatland, 2000; Lepreti et al., 2001) that a power law $f \sim \Delta t^{-3}$ should be observed for the waiting times. Although a hypothetical power law should show a scaling exponent far from the result we obtained using the real dust storms dataset, to dispel any doubts we test, as a zeroth-order hypothesis, whether or not an approach based on the occurrence of a local Poisson process can describe the data.

Let us suppose that the local dust cloud rate is not constant, without invoking any specific probability distribution, and let us conjecture that events are Poissonian, with a density that varies locally in time. Then we prove that this conjecture must be rejected when the real data are taken into account. Let us introduce the minimum local waiting time $\delta t_i = \min\{t_{i+1}-t_i, t_i-t_{i-1}\}$, and let $\delta\tau_i$ be either $\delta\tau_i = t_{i+2}-t_{i+1}$ if $\delta t_i = t_{i+1}-t_i$, or $\delta\tau_i = t_{i-1}-t_{i-2}$ if $\delta t_i = t_i-t_{i-1}$. According to a local Poisson hypothesis, both $\delta t_i$ and $\delta\tau_i$ are identically distributed with a Poisson exponential distribution, namely $f(\delta t_i) = 2\lambda_i \exp(-2\lambda_i \delta t_i)$ and $f(\delta\tau_i) = \lambda_i \exp(-\lambda_i \delta\tau_i)$, respectively, being $\lambda_i$ the local rate. Under the above-mentioned null hypothesis, it is easy to show that the stochastic variable $H = 2\delta t_i/(2\delta t_i + \delta\tau_i)$ is uniformly distributed in $[0, 1]$, that is $H$ has a cumulative distribution that is independent of the local rate $\lambda_i$ and decays linearly, namely (Lepreti et al., 2001; Sorriso-Valvo and et al, 2007)

$$f(H \geq h) \equiv \int_h^\infty f(H)dH = \int_0^\infty dx\, 2\lambda\, e^{-2\lambda x} \int_0^{g(h,x)} dy\, \lambda\, e^{-\lambda y} = 1-h \tag{3}$$

where f(H) is the probability distribution function of $H$ and $g(h,x) = 2x[(1/h)-1]$. This means that, under the hypothesis of local Poissonian events with a continuous distribution for the rate $\lambda$, the stochastic variable $h$ is uniformly distributed with average $1/2$. If the process is characterized by clusters or voids, $h$ will be typically greater or lesser than $1/2$, respectively.

We investigate the null Poissonian hypothesis by calculating $f(H \geq h)$ for all the stations in the dataset and on all time scales. We collect the local values of $h$, whose cumulative distribution are shown in Figure 6 for the stations reported in Figure 1. Even if at large time scales the system becomes Poissonian, the non poissonian nature of the phenomenon survives in the cumulative distribution of $h$. A departure from the local Poisson hypothesis is in fact observed for all the datasets, as the relation (3) is not satisfied, moreover the phenomenon of consecutive dust storms is mainly made by clusters with respect to voids. Namely, the null hypothesis that dust storms are generated by a stochastic Poisson process, with a variable rate of occurrence, must be rejected. The power law statistics we found mean that a non-Poisson process is at work at small time scales, we can claim that the presence of hidden correlations between consecutive inter-seasonal dust storms is a genuine interesting physical phenomenon of the process at hand.



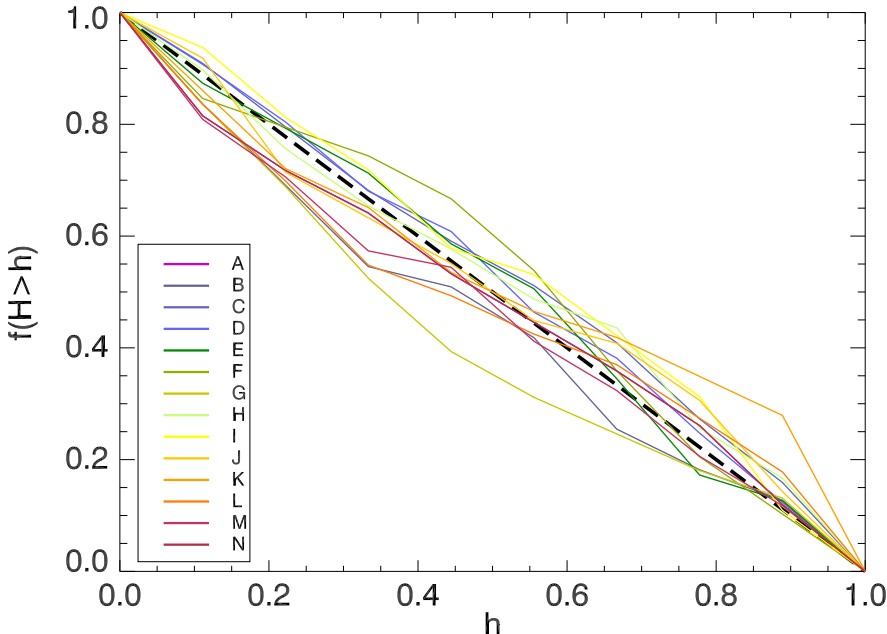

**Figure 6.** Cumulative probability distribution function $f(H \geq h)$ as a function of $h$, for the datasets reported in Figure 1 and Table 1. The dashed line, which is shown as a reference, refers to a Poisson process. Different colours indicate different stations (see Table 1).

## 5 Conclusions

In this paper, we investigated the teleconnection between Northern Africa and the western Mediterranean Basin, through the statistical behaviour of SDO. The teleconnection may be explained by the physical phenomenon that drives the dust clouds toward the western Mediterranean basin, identifiable in the atmospheric circulation patterns (Salvador et al., 2014; Gkikas et al., 2015; Fernandes and Fragoso, 2021; Salvador et al., 2022).

We analysed the Saharan Dust intrusion seasonality depending on the synoptic conditions, which were clustered into 4 different atmospheric circulations, as found in previous studies (Salvador et al., 2014; Fernandes and Fragoso, 2021), although in other works a greater number of clusters was found (Gkikas et al., 2015; Salvador et al., 2022). We highlighted two main clusters, representing the Azores Current and the African anticyclone, favouring the SDO events in the western Mediterranean basin. Except for summer, no significant differences in cluster distributions were found in the other seasons. Indeed, in winter Cluster 1 was the most frequent but it was not significantly linkable to Dust transport, while two minor atmospheric circulations (i.e., Cluster 2 and Cluster 4), inducing the SDO, were found. In spring, the atmospheric conditions of Cluster 1 became less frequent, and the two prevalent clusters were Cluster 2 and Cluster 4 able to transport Saharan Dust frequently. Summer represented a break point with the atmospheric patterns, with a net prevalence of the Azores Current (i.e., Cluster 2) favouring the Dust intrusions. Finally, during autumn the atmospheric pattern changed again and the main synoptic condition inducing



SDO events was given by the African anticyclone (i.e., Cluster 4), followed by the Azores Current (i.e., Cluster 2). Despite little variations in the dominance of main clusters, from the synoptic point of view, the atmospheric conditions in autumn, winter, and spring, can be considered very similar, whereas a very different behaviour was found in summer.

     By looking at basic statistics on persistence times $\tau$ between consecutive Saharan Dust storms that hit the western Mediterranean Basin, we found that the cumulative distribution function of persistence times $f(\Delta t \geq \tau)$ is described by a well-defined

power law with an exponential cut-off for large $\Delta t$. This is a novel and interesting result because it indicates that consecutive dust storms on periods lesser than $50$ days are correlated, thus representing a kind of proxy evidencing a general teleconnection between the Saharan region and the western Mediterranean Basin (Littmann, 1989; Feldstein and Franzke, 2017; Horenko et al., 2017). The characteristic values we found through a fit on the data, namely $\Delta t_0 \simeq 3$ days, and $\gamma^{-1} \simeq 118$ days, represent the basic time scales of the phenomenon. Indeed, $\Delta t_0$, on average, results to be of the order of the classical duration of

each dust storm, while the decay rate $\gamma^{-1}$ represents a kind of memory time, namely after roughly $118$ days the system loses memory of the presence of previous dust storms. The seasonal statistical behaviour highlights that SDO events occurring from autumn to spring, showing a similar distribution of atmospheric circulations, are characterized by a long memory time, while in summer, when only one cluster is very frequent and prevalent, SDO events become early randomly and the memory time is reduced. These results bring to mind that summer events are frequent but driven by only a synoptic circulation, but when the

atmospheric conditions change, the SDO events are no more correlated. On the other hand, similar atmospheric conditions of the other seasons induce a longer memory time.

     The scaling exponent of the power law (3), $\mu \simeq 1$, deserves a separate discussion. This is a peculiar value since in nature a power law with a unitary exponent is a classical phenomenon describing a pink-noise stochastic process, which has been found in a wide range of physical situations, even though its origin is not yet fully understood (Bak, 2013). The presence of a

pink noise describing correlations between consecutive stochastic events is clear evidence of a teleconnection that affects the European environment on long-time scales.

*Code and data availability.* The PM10 datasets used in this work are freely available and can be downloaded at https://www.eea.europa.eu/en. The NAAPS (Navy Aerosol Analysis and Prediction System) Global Aerosol Model maps are available http://www.nrlmry.navy.mil/aerosol. The HYSPLIT (Hybrid Single-Particle Lagrangian Integrated Trajectory) model driven by Reanalysis set as meteorological

conditions is available at https://www.ready.noaa.gov/hypub-bin/trajasrc.pl.

*Author contributions.* V. Carbone and F. L. conceived the statistical analysis and analysed the results, V. Capparelli and J. C. organized and analysed the datasets and the results, A. S. and G. M. analysed the results. All authors reviewed the manuscript.

*Competing interests.* Authors declare no competing interests



*Acknowledgements.* Jessica Castagna acknowledges support from the Next Generation EU - Italian NRRP, Mission 4, Component 2, In-
240 vestment 1.5, call for the creation and strengthening of 'Innovation Ecosystems', building 'Territorial R&D Leaders' (Directorial Decree
n. 2021/3277) - project Tech4You - Technologies for climate change adaptation and quality of life improvement, n. ECS0000009. This
work reflects only the authors' views and opinions, neither the Ministry for University and Research nor the European Commission can be
considered responsible for them. The authors gratefully acknowledge the NOAA Air Resources Laboratory (ARL) for the provision of the
HYSPLIT transport and dispersion model and READY website (https://www.ready.noaa.gov) used in this publication.



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
