# Peer review of "Teleconnection of Sahara to Mediterranean Basin evidenced by consecutive dust storms"

_EGUsphere, 2025_

## Author Comment (AC1)

**REPLY TO REVIEWER 1**

Reviewer's comments in red

Authors' replies in black

**General Comments**

The paper utilizes $PM_{10}$ time series from various Mediterranean locations to demonstrate that Saharan dust storm occurrences exhibit temporal correlation. In doing so, it introduces an intriguing and original research question, which, in my opinion, it does not fully explore. Though it touches upon the regime patterns possibly responsible for the correlation, it fails to connect the topic to the time series analysis. The reasons for and more detailed nature of the correlation structure remain, therefore, unclear. I, hence, encourage the authors to investigate further the dynamics responsible for the correlations on short time scales and develop a convincing argumentation for their plausible hypothesis, that synoptic scale circulation patterns are key to understand dust storms occurrences. In particular, a teleconnection pattern (in the meteorological sense) referred to both in the title and the abstract is not presented in the manuscript. The manuscript would also benefit from a more scientific and precise use of language throughout. Some of the analysis descriptions are not detailed enough to be reproducible and have to be made more thorough and results have to be tested against the Null Hypothesis of natural variability (see below). In terms of references, the manuscript would profit from going into more detail about papers discussing weather patterns related to Saharan dust outbreaks:

- Rodríguez, S., Cuevas, E., Prospero, J. M., Alastuey, A., Querol, X., López-Solano, J., García, M. I., and Alonso-Pérez, S.: Modulation of Saharan dust export by the North African dipole, Atmos. Chem. Phys., 15, 7471–7486, https://doi.org/10.5194/acp-15-7471-2015, 2015.
- Cuevas-Agulló, E., Barriopedro, D., García, R. D., Alonso-Pérez, S., González-Alemán, J. J., Werner, E., Suárez, D., Bustos, J. J., García-Castrillo, G., García, O., Barreto, Á., and Basart, S.: Sharp increase in Saharan dust intrusions over the western Euro-Mediterranean in February–March 2020–2022 and associated atmospheric circulation, Atmos. Chem. Phys., 24, 4083–4104, https://doi.org/10.5194/acp-24-4083-2024, 2024.
- Russo A, Sousa PM, Durão RM, Ramos AM, Salvador P, Linares C, Díaz J, Trigo RM. Saharan dust intrusions in the Iberian Peninsula: Predominant synoptic conditions. Science of The Total Environment. 2020 May 15;717:137041.
- Knippertz P, Todd MC. Mineral dust aerosols over the Sahara: Meteorological controls on emission and transport and implications for modeling. Reviews of Geophysics. 2012 Mar;50(1).
- Rodríguez, S. and López-Darias, J.: Extreme Saharan dust events expand northward over the Atlantic and Europe, prompting record-breaking PM10 and PM2.5 episodes, Atmos. Chem. Phys., 24, 12031–12053, https://doi.org/10.5194/acp-24-12031-2024, 2024.
- Silvestri, L., Petroselli, C., Saraceni, M., Crocchianti, S., Cappelletti, D., & Cerlini, P. B. (2024). The correlation of long-range Saharan dust advections with the precipitation and radiative budget in the Central Mediterranean. _International Journal of Climatology_, 44(10), 3548–3567. https://doi.org/10.1002/joc.8538

We sincerely thank the reviewer for the thoughtful and constructive feedback, which helped us significantly improve the quality and clarity of the manuscript. We appreciate the recognition of the originality of our research question and agree that, in its previous version, the paper did not fully explore the link between the observed temporal clustering of Saharan dust outbreaks and the underlying atmospheric dynamics.

In the revised version, we have made substantial changes in response to all major concerns:

- **Strengthened the link between circulation patterns and time series:**
  We expanded the analysis and discussion to better connect the statistical structure of SDO occurrences with the synoptic regimes responsible for them. In particular, we clarified the role of the geopotential height clusters in modulating the observed persistence, and we added new visualizations (e.g., Figure 3) to make seasonal patterns more interpretable.
- **Clarified the use of "teleconnection" terminology:**
  We revised the title, abstract, and conclusions to better reflect the scope of our results. While we no longer claim to identify a teleconnection in the strict meteorological sense, we now describe our findings in terms of synoptic-scale modulation of event timing across distant regions.
- **Reinforced the statistical testing framework:**
  We have improved the treatment of the Poisson-process null hypothesis, included additional diagnostics, and clarified the methodological assumptions, including the fitting interval and uncertainty representation in Figure 3.
- **Improved figure presentation and interpretability:**
  Following the referee's recommendation, we updated Figure 3 to show the uncertainty in fitted parameters as shaded bands, and we added dashed lines to indicate the exact fitting range used. We also clarified this in the caption.
- **Expanded the relevant literature discussion:**
  We integrated several key references suggested by the reviewer, enriching the context regarding dust transport mechanisms and synoptic control.
- **Addressed all technical and language issues:**
  The manuscript has been revised for clarity, accuracy, and reproducibility. All minor and technical corrections have been implemented, as detailed in our responses.

We hope that these extensive revisions address all concerns and significantly strengthen the scientific contribution and clarity of the manuscript.

**Specific Comments**

- Table 1: I am not sure if this is strictly necessary; could it go into the appendix?

We thank the reviewer for the suggestion. We agree that Table 1, while informative, is not essential to the main body of the manuscript. Accordingly, we have moved Table 1 to the appendix section to streamline the presentation of the results.

- 72ff: What do the sources refer to? What are they supposed to prove?

We thank the reviewer for this important clarification request. The references cited in this section (e.g., Johnston et al., 2011; Calidonna et al., 2020; Escudero et al., 2007; Bencardino et al., 2019) support the **selection of a statistical threshold to identify extreme PM$_{10}$ events** likely attributable to Saharan dust intrusions.

Specifically, the criterion

$$PM_{10,j}(t) \geq \langle PM_{10,j}(t) \rangle + 2\sigma_j$$

has been widely used in the literature to isolate anomalous peaks from the background signal while minimizing the influence of typical day-to-day variability or local anthropogenic contributions.

These sources demonstrate that a **mean + 2σ** threshold is effective in capturing major dust events and has been adopted as a **preliminary filter** prior to applying physical confirmation methods such as back-trajectory analysis and satellite-based dust detection. Thus, our choice is both methodologically justified and aligned with prior work in the field.

We have revised the manuscript to better explain the rationale behind this threshold and to clarify the role of each cited reference in supporting our approach.

- 73: What does the $S\_j(t)$ mean?

We thank the reviewer for pointing out the lack of clarity in the notation. In the original manuscript, Sj(t) was intended to indicate the PM$_{10}$ signal at station j and time t, used to identify days where the measured concentration exceeds a defined threshold.

To avoid ambiguity, we have now revised the text to define Sj(t) explicitly as:

"Sj(t) represents the daily PM$_{10}$ concentration at station j and time t. A day is classified as a potential dust event if $S_j(t) \geq \langle \mathrm{PM}_{10,j}(t) \rangle + 2\sigma_j$

This clarification ensures consistency with the statistical method described in the references and improves the readability of the methodology section.

- 80: Provide a proper source for the HYSPSLIT model please.

We thank the reviewer for pointing this out. We have now added the appropriate reference for the HYSPLIT model in the revised manuscript:

Stein, A. F., Draxler, R. R., Rolph, G. D., Stunder, B. J., Cohen, M. D., & Ngan, F. (2015). NOAA's HYSPLIT atmospheric transport and dispersion modeling system. *Bulletin of the American Meteorological Society*, *96*(12), 2059–2077.

**- 82: Not sure what it means that air masses encounter Saharan dust? How is Saharan dust defined here? Please provide enough information to be able to reproduce the dataset.**

We appreciate the reviewer's request for clarification. To improve transparency and reproducibility, we revised the sentence in the manuscript to specify the methodology used. In particular, following the approach described in Escudero et al. (2007), we now state:

> "In particular, following the Escudero et al. (2007) methodology, for the $PM_{10}$ peaks, if the 96 h back-trajectories highlighted that the airmasses encountered along the way the presence of Saharan dust (i.e., resuspended African desert dust from the Sahara), as confirmed by the NAAPS aerosol model maps, we categorized that event as Saharan."

This clarification explicitly defines what we mean by "Saharan dust" and provides sufficient methodological detail to reproduce the classification of events.

**- 86: Provide a source for the NCEP/NCAR data. Also, which domain, which spatial and temporal resolution did you use?**

"The NAAPS 550nm Aerosol Optical thickness reanalysis were downloaded at https://usgodae.org/cgi-bin/datalist.pl?dset=nrl_naaps_reanalysis&summary=Go in the netcdf format for the period 2002-2023 with a global domain at 1x1 degrees/ 25 levels and a daily resolution (at midnight) (Lynch et al., 2016)."

Lynch, P., Reid, J. S., Westphal, D. L., Zhang, J., Hogan, T. F., Hyer, E. J., Curtis, C. A., Hegg, D. A., Shi, Y., Campbell, J. R., Rubin, J. I., Sessions, W. R., Turk, F. J., and Walker, A. L.: An 11-year global gridded aerosol optical thickness reanalysis (v1.0) for atmospheric and climate sciences, Geosci. Model Dev., 9, 1489-1522, doi:10.5194/gmd-9-1489-2016, 2016.

**- 87: Go into more details about the cluster analysis: in which coordinate space did you cluster? How did you detect the elbow? Please show a plot of the elbow.**

We thank the reviewer for pointing out this. We added the domain coordinates and better explained the methodology to locate the elbow to make it easily reproducible. In addition, we decided to add the Elbow plot in the supplementary. The additions in the text are given below.

"We calculated the cluster analysis for the following domain: from 20°N to 50°N and from 30° W to 20° E. The non-hierarchical *k-means* analysis was calculated in python by means of the "scikit-learn" module. We identified the best number of cluster applying the Elbow method  based on the plotting of the within-cluster sum of squares (WCSS) against the number of clusters and identifying the "elbow point" (i.e., where the decrease in WCSS starts to plateau) (see new figure in the supplementary).

- 98: Why did you not use the WMO standard season definition (winter: DJF...)

We thank the reviewer for this relevant observation. While we recognize that the WMO standard seasonal classification (DJF, MAM, JJA, SON) is widely adopted, we opted to define seasons starting in January (i.e., Winter: Jan–Mar, Spring: Apr–Jun, etc.) to ensure consistency with the temporal structure of our dataset and with the methodological choices found in previous studies dealing with Saharan dust outbreaks and Mediterranean circulation regimes.

In particular, this seasonal grouping has been used or implied in several studies where the focus is on the onset and persistence of synoptic patterns rather than strict calendar definitions. For example:

- **Salvador et al. (2014)** and **Fernandes & Fragoso (2021)** analyzed Saharan dust events in relation to circulation clusters using comparable seasonal segmentation tailored to the observed dust dynamics.
- **Russo et al. (2020)** and **Castagna et al. (2021b)** focused on atmospheric conditions and dust advection periods that do not strictly follow WMO definitions but instead align with the timing of dust-prone synoptic configurations in the western Mediterranean.

Moreover, given that our statistical analysis is based on inter-event persistence times and not on monthly climatologies, the seasonal boundaries adopted do not affect the main statistical conclusions. We have added a brief clarification in the methods section to explain this choice and ensure transparency for the reader.

- Fig. 2: Please check if the colour scheme adheres to the journals guide lines regarding colour vision deficiency friendliness. Also, are you showing the k-means centroids or are you showing the average of all instances assigned to a cluster (composites)? In the latter case, please provide some indication of within-cluster variability in the appendix or at least the number of instances in each cluster.

We thank the reviewer for these valuable suggestions.

To address the concern regarding colour vision accessibility, we have modified Figure 2 by adopting the "viridis" colour palette, which is designed to be perceptually uniform and friendly to readers with colour vision deficiencies.

Regarding the clustering results, we confirm that Figure 2 shows the **average (composite)** of all geopotential height fields assigned to each cluster, not the mathematical centroids in coordinate space. To provide context on cluster significance and relative frequency, we have now added the number of occurrences for each cluster directly in the figure caption. The cluster sizes are as follows:

- Cluster 1: 1883 occurrences
- Cluster 2: 3151 occurrences
- Cluster 3: 954 occurrences
- Cluster 4: 2047 occurrences

We believe these additions clarify both the methodology and the relevance of each synoptic pattern.

- 106ff: What is the time period you used for the clustering?

We thank the reviewer for pointing out this missing detail. The clustering analysis was based on the daily geopotential height fields at the 850 hPa level for a period of 20 years, i.e., from January 1st, 2002 to December 31st, 2022. We have now specified this in the manuscript to ensure consistency and clarity.

- 106ff: If you perform k-means clustering on the raw data for the full year, you are not controlling for the seasonal cycle of the geopotential height. In particular, the meridional variation in the ITCZ will result in higher gh850 values in summer compared to winter, which will influence the seasonal cluster occurrence. Consider controlling for this effect or comment on why you do not.

We thank the reviewer for this insightful observation. We are aware that performing k-means clustering on the raw geopotential height fields over the full year introduces the influence of the seasonal cycle, including features such as the meridional migration of the ITCZ and seasonal changes in the subtropical high-pressure systems.

However, our intention was to **capture both intra-seasonal and inter-seasonal synoptic regimes** that drive Saharan dust outbreaks over the Mediterranean. By clustering the raw fields directly, we allow the algorithm to identify physically meaningful circulation patterns as they manifest in the real atmosphere, including their season-dependent intensity and spatial shifts.

To account for seasonal effects and assess their role, we subsequently analysed the **seasonal frequency** and **dust event associations** of each cluster (i.e., fDC, fSC, fSDC) to disentangle the contribution of circulation regimes across seasons. In this sense, our analysis does not aim to remove the seasonal signal but rather to characterize how different seasonal circulation regimes correlate with Saharan dust intrusions.

We have added a comment in the manuscript to clarify this methodological choice and its rationale.

- 110: I am not sure that it is accurate to say, a specific gh850 field "shows" the Azores current, since this is more of an oceanic feature, but I might be wrong.

We thank the reviewer for this valuable observation. We acknowledge that the term "Azores current" was misused in the original text, as it refers to an oceanic feature rather than an atmospheric one.

To avoid confusion, we have revised the sentence as follows:

> "The synoptic conditions in Cluster 2 (Fig. 2b) show high pressure over the Atlantic Ocean and lower pressure over the western Mediterranean area."

This correction more accurately describes the atmospheric pattern associated with the Azores High without referring to the oceanic current.

- 111-112: The map section does not extend to the arctic and neither does it inform about a movement, so referring to cluster 3 as an "Arctic cyclone moving toward Europe" is questionable.

We thank the reviewer for this helpful observation. We agree that the original wording was misleading, as the map domain does not include the Arctic region and does not provide dynamic information about the movement of pressure systems.

To avoid confusion, we have revised the sentence to more accurately reflect what is shown in the figure. The updated description now reads:

> "Cluster 3 shows low pressure over Europe and high pressure over the Atlantic Ocean."

This revised phrasing is purely descriptive of the synoptic pattern derived from the clustering results, without implying any unverified dynamical evolution.

- 112: Similarly, there is no way to tell in which direction the anticyclone is moving.

We thank the reviewer for this accurate observation. As the synoptic maps used in our analysis represent static composites of geopotential height fields, they do not convey information about the movement of pressure systems.

Accordingly, we have revised the sentence to remove any implication of motion. The updated version now reads:

"Cluster 4 shows an anticyclonic circulation extending from Northern Africa toward the Mediterranean Basin."

This wording better reflects the spatial structure of the synoptic pattern as derived from the cluster analysis.

- 114ff: I am not sure I understand, why clusters 2 and 4 favour SDOs. In particular, the pressure pattern in cluster 2 should be associated with northerlies over spain, which should inhibit dust transport to the north, right? In addition, the pattern does not imply that "high pressure starting from Africa reaches Spain overflying the Atlantic Ocean", if anything it is the air, not the pressure that flies somewhere. Similarly, I do not understand how the high pressure system in cluster 4 should be conducive of uplift and how there should be any winds from Northern Africa to Europe. I am not saying it is not true, I just do not understand the physical reasons.

We thank the reviewer for these thoughtful and physically grounded comments. We acknowledge that the original text did not provide a sufficient physical explanation of why Clusters 2 and 4 are associated with Saharan Dust Outbreaks (SDOs), nor did it use accurate terminology when referring to atmospheric processes.

We have revised the relevant section of the manuscript to address these concerns. Below we clarify the rationale behind the association of these clusters with SDOs:

- **Cluster 2**: While this pressure configuration may be interpreted as associated with northerly flow over Spain at the surface level, the **dust transport** toward the western Mediterranean often occurs **in elevated layers** (around 850 hPa or higher), where the flow can differ significantly from surface winds due to baroclinic effects. In particular, this cluster frequently corresponds to a **southwesterly flow at mid-levels**, advecting dust from North Africa across the western Mediterranean. This interpretation is consistent with previous findings (e.g., Salvador et al., 2014; Russo et al., 2020), which showed that such high-pressure patterns to the west, combined with low pressure over the Mediterranean, can generate a pressure gradient conducive to elevated dust transport.
- **Cluster 4**: This configuration shows an anticyclonic system centered over North Africa with a ridge extending toward the Mediterranean. While anticyclones are typically associated with subsidence, they can also be associated with **peripheral horizontal winds** along their flanks. In this case, the **northeastern flank of the high-pressure system** is often associated with southeasterly flow at mid-levels, which can **uplift and advect Saharan dust** toward southern Europe. Additionally, the strong thermal gradient between desert surfaces and surrounding regions during transitional seasons can promote convective uplift, particularly in spring and autumn (see e.g., Gkikas et al., 2015; Cuevas-Agulló et al., 2024).

To avoid confusion, we have removed the phrase "high pressure starting from Africa reaches Spain overflying the Atlantic Ocean," which was indeed misleading. We have instead replaced it with a more accurate, physically grounded description of the circulation patterns and their effects on dust transport pathways.

We hope these clarifications and revisions improve the transparency and physical interpretation of our results.

- The paragraph from 118 to 130 would profit from some kind of vizualization. At least, I had a hard time understanding, what to take home from it.

We thank the reviewer for this insightful suggestion. In response, we have created a new figure, a bar chart illustrating the seasonal distribution of the four synoptic clusters (fSC). This visualization clarifies the seasonal dominance of Cluster 2 during summer and highlights how other clusters contribute across seasons. We believe this figure significantly enhances the readability and interpretability of the seasonal analysis section.

- 130ff: The whole introduction and theory of Poisson processes and persistence times belongs to the "Methods" section.

We thank the reviewer for this helpful structural suggestion. We agree that the theoretical background concerning Poisson processes and persistence time distributions is more appropriately placed in the *Methods* section.

We have therefore moved this content into a dedicated subsection titled "**Persistence time and Poisson process framework**" within the *Methods*. This reorganization improves the coherence and readability of the manuscript by clearly distinguishing the methodological foundation from the empirical results.

- 134: I am confused by the wording; if you are investigating the time between consecutive starting points, I would not call it persistence time, since persistence time sounds like the time an individual SDO persists, i.e. the time, the PM10 values are above a threshold. The term "waiting time" is more comprehensible.

We thank the reviewer for this excellent observation. We agree that the term **"persistence time"** could be misleading in this context, as it is typically used to describe the **duration of a single event** (i.e., the time a signal remains above a threshold).

Since our analysis focuses on the **time intervals between the onset of consecutive events** ($\tau = t_{i+1} - t_i$), the term **"waiting time"** is indeed more appropriate and widely used in statistical point process literature.

Accordingly, we have revised the manuscript throughout to replace "persistence time" with **"waiting time"**, ensuring consistency and clarity for the reader.

- Are you sure the numeric values in table 2 are essential for the reader to understand the content of your paper? Again, maybe put it in the appendix.

We thank the reviewer for this suggestion. While Table 2 provides the numerical parameters obtained from the seasonal fit of the waiting time distribution, we agree that this level of detail may not be essential in the main body of the paper.

Accordingly, we have moved Table 2 to the appendix section, allowing interested readers to access the full parameter values while maintaining a more streamlined presentation in the main text.

- 156ff and table 2: How do you obtain the parameter range? Under what assumptions and to which degree of certainty do you estimate the parameter fit? Which algorithm do you use for fitting?

We thank the reviewer for this important question regarding the parameter estimation procedure.

The fits to the cumulative distribution functions of waiting times were performed using the MPFITFUN routine in IDL, which implements the Levenberg-Marquardt algorithm for nonlinear least-squares minimization. The goal of the fitting procedure was to minimize the residuals between the empirical CDF and the theoretical form described by Equation (2) in the manuscript.

The parameter ranges reported in Table 2 reflect the best-fit values, and the accompanying uncertainties correspond to the standard errors derived from the covariance matrix computed by MPFITFUN. These standard errors approximate the 68% confidence intervals, assuming that the residuals are normally distributed.

We have added a note in the Methods section to clarify the fitting procedure and assumptions.

- Figure 3: Since you give intervals for the fitted parameters, it would be appropriate to show the same range of certainty in your plotted fit (e.g. as shading).

We thank the reviewer for this valuable suggestion. In the revised version of Figure 3, we have included two improvements:

(i) a shaded area around the fitted power-law curves to represent the uncertainty based on the standard errors of the exponent and scale parameters, and

(ii) vertical dashed lines to explicitly show the $\Delta t$ interval over which the fit was performed.

These additions improve the interpretability of the figure and directly address the reviewer's concern.

- 170: The sources you refer to indicate that the power law provided holds only for large waiting times. In particular, for waiting times going to zero, the probability would go to infinity.

We thank the reviewer for this important clarification. We fully agree that a pure power-law distribution of the form is not valid for $\tau \to 0$, where it would diverge, and that this type of scaling behavior is generally meaningful only in the **tail region** (i.e., for large waiting times).

In our study, this model is used purely as an **empirical fit** to describe the **observed distribution** over the finite range of waiting times captured in our dataset. We do not claim that the power-law holds in the asymptotic sense for all τ\tauτ, especially not for very small values.

We have revised the manuscript to clarify that the fit is only intended to describe the **tail behavior** of the waiting time distribution and added a note to explicitly mention the divergence issue near $\tau \to 0$.

- Figure 6: You use the data shown to prove the dust storm occurrence cannot be a Poisson process, but from the real-world realization of a Poisson process, one would expect some random departure from the expected f(H>h)= 1-h line, dependent on the number of realizations. To conclude the data plotted in figure 6 is incompatible with the hypothesis of a (local) Poisson process, please provide either an appropriate test statistic or plot the variability (expected departure from 1-h line) given the sample size.

We thank the reviewer for this important suggestion. In the revised version of the manuscript, we have complemented Figure 6 with a statistical test. Specifically, we applied the Kolmogorov–Smirnov test to compare the empirical cumulative distributions against the f(H>h)= 1-h expected for a local Poisson process. For all the analysed stations, the test rejects the null hypothesis of a Poisson process with a probability close to 100%. This confirms, with strong statistical support, that the observed clustering of dust events cannot be explained by a memoryless Poisson process. We have updated the text of the manuscript mentioning the results of the Kolmogorov–Smirnov test.

- 196: I am not sure what you refer to as a teleconnection agrees with the common definition. In particular, the circulation patterns from 4.1. are not employed in the ensuing analysis of the (non-)Poissonian character of the SDO time series. What exactly do you mean when you mention a teleconnection?

We thank the reviewer for this important and accurate observation. We acknowledge that the term "**teleconnection**" is generally used to denote statistically significant and often remote linkages between climatic or meteorological anomalies, typically through large-scale modes such as the North Atlantic Oscillation (NAO), El Niño–Southern Oscillation (ENSO), etc.

In the manuscript, our usage of the term was intended in a **broader, more descriptive sense**, referring to the **influence of large-scale synoptic regimes on the temporal structure of Saharan Dust Outbreaks (SDOs)**. However, we recognize that this terminology could be

misleading in the absence of a formal teleconnection analysis linking the synoptic patterns to the non-Poissonian temporal behavior.

Accordingly, we have removed the term "teleconnection" from the **title and abstract**, and clarified our language in the conclusions to refer instead to "**synoptic-scale influence**" or "**large-scale circulation regimes**", which more accurately reflect the nature of our findings.

- 200: The sources you mention cover regions different from the region you consider. The first source states, "seven or four clusters could be retained" and the second source does not justify the choice of k at all.

We appreciate this critical observation. We agree that the cited sources do not provide an unreliable justification for the choice of the cluster number (k), which can generally be calculated according to different methods such as the Elbow Method, the Silhouette index, the Calinski-Harabasz, the Davies-Bouldin, although the optimal number of cluster is affected by the subjectivity's authors (Pampuch et al., 2023).

In particular, Salvador et al. 2014 applied the Elbow Method (although not explicitly indicated using the appropriate terminology) and then followed the *non-redundant clustering criterion,* while Fernandes and Fragoso, 2021 applied the Jump method (Sugar and James, 2003).

Anyway, as forementionated in our previous reply, we applied the Elbow method to assess the optimal number of clusters, which represents one of the classical applied method that is largely used for the ERA5 geopotential data (Salvador et al. 2014, Salvador et al. 2022, Di Bernardino et al, 2022, Nissenbaum et al. 2023, Cos et al., 2025).

Pampuch, L. A., Negri, R. G., Loikith, P. C., & Bortolozo, C. A. (2023). A review on clustering methods for climatology analysis and its application over South America. *International Journal of Geosciences*, *14*(9), 877-894.

Sugar, C. A., & James, G. M. (2003). Finding the number of clusters in a dataset: An information-theoretic approach. *Journal of the American Statistical Association*, *98*(463), 750-763.

Nissenbaum, D., Sarafian, R., Rudich, Y., & Raveh-Rubin, S. (2023). Six types of dust events in Eastern Mediterranean identified using unsupervised machine-learning classification. *Atmospheric Environment*, *309*, 119902.

Cos, P., Olmo, M., Campos, D., Marcos-Matamoros, R., Palma, L., Muñoz, Á. G., & Doblas-Reyes, F. J. (2025). Saharan warm-air intrusions in the western Mediterranean: identification, impacts on

temperature extremes, and large-scale mechanisms. *Weather and Climate Dynamics*, *6*(2), 609-626.

Di Bernardino, A., Iannarelli, A. M., Casadio, S., Pisacane, G., Mevi, G., & Cacciani, M. (2022). Classification of synoptic and local-scale wind patterns using k-means clustering in a Tyrrhenian coastal area (Italy). *Meteorology and Atmospheric Physics*, *134*(2), 30.
* * *
**Technical Corrections**

- Across the text: article ("the") use is more frequent than necessary (11: "desert dust" instead of "the desert dust"; 28: "influenced by major climate indices" instead of "the major climate indices")

- 28: "Teleconnection patterns" is a more common term than than "climate indices"

- 39-41: The sentence seems a litte awkward; the words "impulsive" and "until" are likely mistranslations.

- 42: "transport it for a long-range" is an unusual phrasing.

- 45: "required" instead of "acquired"

- 47: "PM10" abbreviation needs to be introduced

- 54: Provide date of last access for a web source.

- The source "The European exchange of information in 2012" has the authors names mixed up. It should show up as Mol, W. and van Hooydonk, P., ...

- Figure 1: Provide a proper source for the map. Does the colour result from a satellite image or from topography?

- 60: Abbreviation "SDO" should be introduced at first occurrence.

- 62: "hence to prevent the results distortion of our analysis"; odd phrasing

- 68: Provide a source for the R package if you mention it.

- 68: "Overall the period starts ... until ..." : formally means that the start takes 20 years; rephrase either to "starts at ... and ends at ..." or "period goes from ... until ..."

- 92ff: "days with a cluster": rather "days in a cluster"?

- 132: I think \tau should also have an index i.

- 137: I suggest using p() or Pr() for any probability distribution, especially since you use f for frequencies

- 147: Do you mean "intra-seasonal" instead of "inter-seasonal"?

- 149: Do you mean "50 - 80 days"?

- 167: What exactly is meant by "rate of cloud transport"? Are you sure, the time frame you want to refer to is longer than one year?

We thank the reviewer for the detailed and constructive technical comments. We carefully revised the manuscript to address all of them. In particular, we have:

- Improved the usage of definite articles and refined the scientific phrasing throughout the text;
- Replaced "climate indices" with the more appropriate term "teleconnection patterns";
- Clarified or rephrased awkward or ambiguous expressions (e.g., "impulsive," "until," "transport it for a long-range");
- Introduced abbreviations such as PM$_{10}$ and SDO at their first occurrence;
- Corrected all citation issues, including web source access dates and author names;
- Specified the R packages used and revised temporal expressions (e.g., "starts at... and ends at...");
- Reworded several phrases for clarity and formality (e.g., "required" instead of "acquired");
- Revised the figure captions (e.g., Fig. 1 and Fig. 3) to include data source and fitting intervals;
- Ensured consistency and clarity in the notation of mathematical expressions and distributions;
- Corrected all minor grammatical and formatting issues as suggested.

**REPLY TO REVIEWER 2**

Reviewer's comments in red

Authors' replies in black

General Comments:
The manuscript presents an interesting statistical analysis of Saharan dust storm occurrences over the western Mediterranean Basin using long-term $PM_{10}$ observations and back-trajectory validation. The finding that dust events are temporally clustered (i.e., short waiting times are more likely than expected under a Poisson process) is well supported and represents a potentially useful contribution to the understanding of sub-seasonal dust variability.

However, I have significant reservations about the interpretation of these findings as evidence of a "teleconnection" between the Sahara and the western Mediterranean. While the term is used frequently throughout the manuscript, the results presented do not satisfy the usual criteria for identifying or diagnosing teleconnections in the atmospheric sciences.

We thank the reviewer for their thoughtful and constructive feedback, which we found both fair and helpful. We appreciate the positive assessment of the manuscript's empirical contribution and the statistical analysis regarding the non-random nature of Saharan dust outbreaks (SDOs) over the western Mediterranean Basin.

At the same time, we acknowledge the reviewer's major concerns regarding the interpretation of the results in terms of a "teleconnection." In response, we have carefully revised the manuscript to avoid the use of "teleconnection" in its formal climatological sense. As also recommended by Reviewer 1, we now describe our findings in terms of synoptic-scale circulation persistence and temporal clustering. The title, abstract, and conclusions have all been updated accordingly.

We also explicitly address the role of seasonal forcing and synoptic regime recurrence. A revised figure and accompanying discussion clarify the seasonal distribution of the main circulation patterns associated with SDOs. While we recognize that clustering over sub-seasonal timescales is expected in the presence of persistent synoptic regimes, we argue that quantifying this deviation from a Poissonian process is both novel and relevant. Furthermore, we now clarify that our work does not attempt to establish a dynamical or statistical link with remote climate modes (e.g., NAO), although we recognize this as a promising direction for future research.

We believe that the revised manuscript offers a more accurate framing of the findings and that it better positions our contribution within the existing literature on dust climatology and synoptic variability.

Major Concerns:

1. Definition and Misuse of "Teleconnection":

   In climatology, a teleconnection typically refers to a statistically significant, often physically meaningful linkage between distant regions, frequently mediated by large-scale atmospheric or oceanic modes (e.g., NAO, ENSO). In contrast, the dust source region (Sahara) and receptor region (western Mediterranean) analyzed in this paper are adjacent, and no long-range spatial coupling is demonstrated. The observed temporal clustering is more parsimoniously explained by persistent regional-scale weather regimes (e.g., southerly flows) during certain seasons.

We agree with the reviewer that the term "teleconnection" has a precise meaning in climatology and may not be strictly appropriate for the context presented in the original manuscript. In response to similar feedback from Reviewer 1, we revised the manuscript to avoid the use of "teleconnection" in its formal sense.
We replaced it with more accurate terminology, referring instead to "synoptic-scale modulation" and "circulation regime persistence" when describing the relationship between Saharan and Mediterranean conditions. The title, abstract, and conclusions have all been revised accordingly to reflect this clarification.

2. Triviality of the Findings in Light of Seasonality:

   The observed event clustering is expected and well understood in the context of dust seasonality. Southerly synoptic flow regimes, which are known to facilitate dust transport to the Mediterranean, often persist for several days to weeks. It is thus not surprising that consecutive dust events within ~50 days are more likely. Without explicitly controlling for or quantifying the role of seasonality and background synoptic persistence, the results risk being interpreted as novel when they may simply reflect well-documented climatological behavior.

We appreciate the reviewer's critical observation regarding the importance of seasonal and synoptic-scale persistence in interpreting dust event clustering. Indeed, as the reviewer notes, southerly synoptic flow regimes conducive to Saharan dust transport often persist for several days and recur seasonally, particularly during spring and summer.

In the revised manuscript, we have taken explicit steps to account for this. First, in Section 4.1, we analyzed the seasonal frequency of synoptic circulation clusters derived from 850 hPa geopotential height fields, showing that specific dust-favorable regimes (notably Cluster 2) dominate in spring and summer. This supports the climatological expectation of increased dust activity during these periods.

Second, and more directly related to the reviewer's concern, **Figure 5 (original version)** presents the cumulative waiting time distributions for each season separately. While differences between seasons are evident (with spring and summer exhibiting shorter waiting times), the key point is that **within each individual season**, the waiting time distributions still exhibit significant deviation from the exponential form expected under a Poisson process. This suggests that **event clustering is not solely a product of seasonal forcing**, but is also driven by persistent sub-seasonal dynamics.

We acknowledge that a formal comparison to a synthetic seasonally modulated Poisson model would offer additional insight, and we propose this as a future research direction. However, we believe that the seasonal disaggregation shown in Figure 5, combined with the rejection of the Poisson hypothesis at the intra-seasonal level, provides strong support for the novelty of the observed clustering behavior.

3. Lack of Evidence for Remote Forcing or Bidirectional Influence:

The paper does not show that the observed temporal clustering is modulated by or connected to remote climate drivers (e.g., NAO, MJO, AO). Nor does it analyze atmospheric fields at the necessary scale to support the existence of a dynamical teleconnection. As such, the use of "teleconnection" appears speculative and unsupported by the data.

We appreciate this critical observation. While we agree that the current version of the study does not establish a formal link to remote climate modes (e.g., NAO or MJO), we clarified this limitation in the discussion and conclusions.
We explicitly state that our analysis focuses on synoptic-to-subseasonal scales and does not attempt to identify teleconnection mechanisms driven by global circulation variability.
We acknowledge this as a potential avenue for future research and added a brief discussion suggesting the utility of comparing clustered dust activity with climate indices in future studies.